# Effectiveness and Safety of Progressive Loading–Motion Style Acupuncture Treatment for Acute Low Back Pain after Traffic Accidents: A Randomized Controlled Trial

**DOI:** 10.3390/healthcare11222939

**Published:** 2023-11-10

**Authors:** Seung-Yoon Hwangbo, Young-Jun Kim, Dong Guk Shin, Sang-Joon An, Hyunjin Choi, Yeonsun Lee, Yoon Jae Lee, Ju Yeon Kim, In-Hyuk Ha

**Affiliations:** 1Bucheon Jaseng Hospital of Korean Medicine, Bucheon-si 14598, Republic of Korea; lead9371@jaseng.org (S.-Y.H.); kimyj_120@jaseng.org (Y.-J.K.); nerodks@jaseng.org (D.G.S.); dkstkdwns10@jaseng.org (S.-J.A.); doctorchj@jaseng.org (H.C.); ewidesun@jaseng.org (Y.L.); 2Jaseng Spine and Joint Research Institute, Jaseng Medical Foundation, Seoul 06110, Republic of Korea; goodsmile@jaseng.org

**Keywords:** progressive loading–motion style acupuncture treatment, exercise therapy, acupuncture, low back pain, randomized controlled trial, Korean traditional medicine

## Abstract

Background: Traffic injuries include acute low back pain (LBP) needing active treatment to prevent chronicity. This two-armed, parallel, assessor-blinded, randomized controlled trial evaluated the effectiveness and safety of progressive loading–motion style acupuncture treatment (PL-MSAT) for acute LBP following traffic accidents. Methods: Based on an effect size of 1.03, 104 participants were recruited and divided in a 1:1 ratio into PL-MAST and control groups using block randomization. Both groups underwent integrative Korean medicine treatment (IKMT) daily; only the PL-MSAT group underwent three PL-MSAT sessions. The outcomes were assessed before and after the treatment sessions and at 1 and 3 months post-discharge. The primary outcome was the difference in the numeric rating scale (NRS) for LBP. The secondary outcomes included a visual analog scale for LBP, leg pain status, the Oswestry disability index, lumbar active range of motion (ROM), quality of life, Patient Global Impression of Change, and Post-Traumatic Stress Disorder Checklist adverse events. Results: In the modified intention-to-treat analysis, 50 and 51 participants were included in the PL-MSAT and control groups. On Day 4, the mean LBP NRS score was 3.67 (3.44–3.90) in the PL-MSAT group, indicating a significantly lower NRS 0.77 (0.44–1.11) compared to 4.44 (4.20–4.68) for the control group (*p* < 0.001). The PL-MSAT group exhibited greater ROM flexion (−5.31; −8.15 to −2.48) and extension (−2.09; −3.39 to −0.80). No significant differences were found for the secondary outcomes and follow-ups. Conclusions: Compared with IKMT alone, PL-MSAT plus IKMT showed significantly better outcomes for reducing pain and increasing the ROM in acute LBP.

## 1. Introduction

Road traffic injuries are a global health problem. The global number of road traffic deaths continues to increase steadily, from 1.15 million in 2000 to 1.35 million in 2018 [1]. Examining a report on car ownership in Korea reveals a 47-fold increase in the number of cars owned by individuals over 30 years, from 530,000 in 1980 to 24.91 million in 2021. The increase is attributable to the growth of the national economy and population, the construction of new roads, and the expansion of existing roads. The number of road traffic accidents (TAs) in Korea has increased by 1.2% annually since 2012, while the number of people suffering road traffic injuries has been increasing by 1.6% annually [2].

The number of patients treated at Korean medicine-related hospitals and clinics for symptoms after road TAs increased by annual averages of 8.65% and 1.30%, respectively, for five years from 2017, while the associated Korean medicine-related medical expenses increased by an annual average of 18.70% [3].

After a road traffic accident, victims may experience various types of musculoskeletal pain, internal bruising, and psychological trauma [4,5]. A nationwide study in the U.S. of more than 21,000 injury claims over two weeks by the Insurance Research Council reported a 58% rate of low back pain (LBP) after a road traffic accident, second only to neck pain (66%) [6]. Recovery from acute LBP after a traffic accident often takes a long time or is never complete; 31% of patients/victims continue to complain of LBP 12 months after the acute injury [7]. Since LBP is a common symptom after a road traffic accident and may develop into chronic pain, patients must receive appropriate treatment in the early stages from the onset. Even when not caused by a traffic accident, LBP is a major cause of limited activity and absenteeism worldwide, and it causes a significant economic burden on individuals, families, communities, industries, and governments [8].

Acupuncture is an important treatment method for TA-caused symptoms in traditional Korean and Chinese medicine; most existing studies on its effects on LBP have reported promising results [9,10,11]. The effectiveness of acupuncture treatment may be further enhanced by applying stimulation through the patient’s movements after needle insertion. Motion style acupuncture treatment (MSAT) is a novel treatment based on acupuncture in which the patient performs active or passive movements with the needle inserted into the acupoints in the affected area [12]. Progressive loading-MSAT (PL-MSAT) combines the MSAT and progressive loading treatment methods. While the needles are inserted in the affected area, the patient is asked to walk a certain distance under the vertical load of sandbags and then actively walk in place (Figure 1A,B).

MSAT has been increasingly applied in Korea and China to enhance the efficacy of traditional acupuncture treatment for patients with musculoskeletal diseases [13,14,15]. The treatment method has been effective for a range of musculoskeletal disorders, including radiating pain in the leg caused by a herniated intervertebral disc (HIVD) in the lumbar spine [16], chronic [17] or acute [18] neck pain, acute whiplash injury [19], and shoulder pain [15].

The PL-MSAT method has been applied for a wide range of symptoms, including acute LBP, back pain, pelvic pain caused by lumbar HIVD, and other lumbar spine-related diseases [20]. However, the limited research to date has only explored the effectiveness and safety of PL-MSAT, resulting in a lack of data on the effectiveness of the therapy for treating acute LBP. Thus, this study aimed to examine the application and effectiveness of PL-MSAT for treating acute LBP.

This randomized controlled trial (RCT) aimed to evaluate the effectiveness of combining integrative Korean medicine treatment (IKMT) and PL-MSAT in reducing pain and improving the functional outcomes and safety of patients with acute LBP caused by TAs.

## 2. Materials and Methods

### 2.1. Study Design and Setting

This study aimed to investigate the effectiveness and safety of PL-MSAT in improving the clinical outcomes of patients with LBP and lumbar dysfunction. A two-arm, parallel, assessor-blinded, randomized controlled study was designed, and 104 participants were recruited at the Bucheon Jaseng Hospital of Korean Medicine, Korea. The Bucheon Jaseng Hospital of Korean Medicine is designated as a specialized traditional Korean medicine hospital for the treatment of spine disorders and it is certified by the Ministry of Health and Welfare, Korea. This RCT was registered on Clinicaltrials.gov (clinical trials ID: NCT05493007).

### 2.2. Participant Timeline

At the first visit, the participants signed an informed consent form (ICF) after a briefing session about the study from a study investigator. Subsequently, the assessor proceeded with the screening procedure according to the inclusion/exclusion criteria to determine the eligibility of the participants. The selected participants were randomized to the PL-MSAT group or control group on the first day of hospitalization (V1).

After completing the screening process, all participants underwent IKMT sessions during the hospitalization period; for those assigned to the PL-MSAT group, PL-MSAT was administered in addition to IKMT on Days 2 (V2), 3 (V3), and 4 (V4) of hospitalization. The participants underwent an assessment of outcomes and tests after screening and following discharge, and they underwent the follow-up process at 1 month and 3 months after discharge (twice in total). Among the outcomes, the numeric rating scale (NRS), visual analog scale (VAS), and range of motion (ROM) were measured before and after the intervention at V2, V3, and V4. The primary endpoint was set as the time of V4. The participants underwent follow-ups 1 month and 3 months after discharge through visits, posts, e-mails, phone calls, or Google survey forms. The schedule of enrollment, interventions, and assessments for participants are presented in Appendix A.

### 2.3. Eligibility

The inclusion/exclusion criteria of the participants are outlined below. 

#### 2.3.1. Inclusion Criteria

The inclusion criteria were (1) male and female adults aged 19–70 years; (2) patients with difficulties in daily living and a need for hospitalization because of acute LBP caused by TAs, without any other traumatic injuries; (3) patients with a Numeric Rating Scale (NRS) for LBP of ≥5; and 4) patients who provided voluntary consent to participate in the RCT and signed the ICF.

#### 2.3.2. Exclusion Criteria

The exclusion criteria were as follows: (1) patients diagnosed with a specific, serious disease that may cause lumbar spinal pain, e.g., malignancies, spondylitis, inflammatory spondylitis, etc.; (2) patients with progressive neurological deficits or severe neurological symptoms; (3) patients who have had spinal surgery or procedures within the last three weeks; (4) patients with pain caused by soft tissue disease, other than spinal pain, e.g., tumors, fibromyalgia, rheumatoid arthritis, gout, etc.; (5) patients with other chronic conditions that may interfere with the interpretation of therapeutic effects or outcomes, e.g., cardiovascular disease, kidney disease, diabetic neuropathy, dementia, epilepsy, etc.; (6) patients taking corticosteroids, immune-suppressive drugs, psychotropic medications, or other drugs that may affect the study outcomes; (7) cases where acupuncture is considered inadequate or unsafe, such as in patients with hemorrhagic disease and receiving anticoagulants, severe diabetes with a risk of infection, or severe cardiovascular disease; (8) patients who are pregnant or planning to conceive; (9) patients with a serious mental illness; (10) patients currently participating in clinical trials other than observational studies without therapeutic intervention; (11) patients incapable of providing informed consent; (12) and other cases of patients whose participation in the trial is deemed problematic as per the clinical judgment of the investigator.

### 2.4. Randomization and Allocation Concealment

For the participants who were determined to be suitable for study participation, a randomization table created with R Studio 1.1.463 (© 2009–2018 R Studio, Inc., Boston, MA, USA) was used for the allocation of an equal number of participants (N = 52, respectively) to the two groups. Block randomization was performed for the generation of random sequences, with the size of one block being randomly set to 2, 4, or 6. The generated results of the randomization were sealed in opaque envelopes and stored in a double-lock cabinet by a third party unrelated to the study. An investigator opened the randomization envelope in front of each participant to assign the participant to one of the two groups. The randomization number assigned to each participant was recorded on an electronic chart.

### 2.5. Blinding

Since the study’s design does not allow for the blinding of the physicians (Korean Medicine Doctor, KMD) and participants, only assessor blinding was applied in this study. The assessors, who remained blinded to the group allocation, assessed the participants in a separate area before the intervention.

### 2.6. Sample Size

The study’s null hypothesis proposed that there would be no difference in the pain outcomes between the experimental (PL-MSAT group) and control groups when IKMT and PL-MSAT were administered in combination or when IKMT alone was administered to participants with acute LBP caused by road TAs. With a significance level of α = 0.05 (two-tailed test) and a type II error (β) of 0.1, the statistical power was set to 90%. A pilot study was conducted to determine the sample size. The mean difference of the NRS was 1.30, with a standard deviation of 1.26. The effect size calculated using these values was 1.03. The correlation with the baseline was calculated to be 0.73, and a minimum of 10 participants were required in each group. Assuming a dropout rate of 20%, 26 participants were required, with 13 in each group. Considering the various subgroup analyses that would be performed in this study, it was decided that 104 participants would be recruited.

### 2.7. Interventions

All interventional procedures of PL-MSAT and IKMT were conducted by KMDs with more than 3 years of clinical experience and who have completed training in standardized Korean medicine and an educational course on PL-MSAT at Bucheon Jaseng Hospital of Korean Medicine. All participants received IKMT during the hospitalization period, and those in the PL-MSAT group underwent PL-MSAT at V2, V3, and V4 (a total of three times).

#### 2.7.1. Control Group: Integrative Korean Medicine Treatment

Participants in the control group received only IKMT daily from V1 until discharge. IKMT consisted of acupuncture, pharmacopuncture, Chuna manual therapy, and herbal medicine.

For the acupuncture treatment, standardized sterile, disposable stainless steel needles (0.25 mm *×* 0.30 mm, Dongbang Medical; Boryeong, Republic of Korea) were inserted into the selected 6–12 acupuncture points among the “essential acupoints” and “optional acupoints” on the bilateral erector muscle of the spine. The essential acupoints are bilateral BL24 (氣海兪, Gihae-su), BL25 (大腸兪, Daejang-su), and BL26 (關元兪, Gwanwon-su). For the optional acupoints, among BL40 (委中, Wijung), SP6 (三陰交, Sameumgyo), BL23 (腎兪, Sin-su), and Hyeopcheok points of 1.5 cm between L3 and L5, a maximum of 6 points were selected according to the clinical judgment of the physician. According to the characteristics of each acupuncture point, perpendicular needling (直刺, chikcha) or oblique insertion (斜刺, saja) was performed, followed by twisting and twirling of the needles to induce the Deqi sensation. The needle retention time was 15 min, and the participants received two sessions of acupuncture treatment daily.

In pharmacopuncture, based on the identification of the pattern of a participant’s constitution and symptoms of the disease, medicinal extracts from specific herbal medicines are injected into disease-related acupuncture points or on the reaction points on the body’s surface [21]. In this study, pharmacopuncture was administered at the time of the acupuncture session considering the conditions of the inpatients, and standardized disposable syringes (1 mL, 29 G × 1/2 syringe, Shinchang Medical, Sungnam, Republic of Korea) were used. According to the identification of the pattern of a participant’s constitution and symptoms of the disease, pharmacopuncture medicine, such as Shinbaro (Jaseng herbal dispensary, Korea), Hwangryunhaedok (Jaseng herbal dispensary, Korea), Joongseongouhyul (Jaseng herbal dispensary, Korea), or bee venom (Jaseng herbal dispensary, Korea), was administered prior to the acupuncture treatment. The injection (0.25 cc) was administered subcutaneously at a depth of 1 cm at the pain sites and the related acupuncture and meridian points. 

Chuna manual therapy is a manually performed therapy in traditional Korean medicine in which the KMD uses their hands, part of their body, or tools (such as the Chuna table) to apply effective stimulation to the participant’s body structures to treat structural or functional problems. The types of Chuna techniques include joint mobilization, joint distraction, myofascial release Chuna, and correction. In this study, Chuna manual therapy was administered once daily for 10 to 15 min during the hospitalization period.

For herbal medicine, decoctions of extracts with mixed formulations of effective medicinal herbs, such as 活血 (activate blood), 理氣 (regulate qi), 補血 (tonify blood), 鎭痛 (analgesic effect), 安神 (tranquilize), and 祛痰 (dispel phlegm), were packed into pouches, and the inpatients were instructed to take the decoction in the morning and afternoon, 30 min after meals.

#### 2.7.2. PL-MSAT Group: Integrative Korean Medicine Treatment and Progressive Loading–Motion Style Acupuncture Treatment

Participants in the PL-MSAT group received IKMT from V1 until discharge, and additionally underwent sessions of PL-MSAT once daily from V2 to V4. Details on the procedure for PL-MSAT administration are displayed in Figure 1.

For PL-MSAT, the participant was first asked to sit in a cross-legged position on a bed or chair. The physician identified GV4 (命門, Myeong-Mun) by palpating between the spinous processes using their thumb. LBP at the lower lumbar spine during palpation indicated GV3 (腰陽關, Yoyang-gwan). Perpendicular needling into GV4 or GV3 was performed, but approximately 3 mm of the body of the needle was not inserted to facilitate its spontaneous insertion when the participant stood up afterward (Figure 1A). With the participant in a cross-legged position with the needle partially inserted, the physician gently rocked the participant’s upper body on either side 15 times (Figure 1B). The participant was subsequently asked to stand with the partially inserted needle, and another needle was inserted into bilateral LR2 (行間, Haeng-gan) (Figure 1(C-1,C-2)). The physician then instructed the participant to walk in place for 15 s (Figure 1D), followed by walking forward for a certain distance. Subsequently, the participant was asked to fold both hands at the elbows, atop which the physician placed one sandbag. The participant was then instructed to walk back and forth twice in a straight line over a distance of 10 m. Subsequently, as the ligaments gained some strength, the physician added another sandbag (Figure 1E), and the participant walked back and forth twice in the same manner. The physician then added another sandbag, totaling three sandbags, and the participant walked back and forth twice (Figure 1F). Thereafter, the cooling-down stage of the PL-MSAT session was started, where the physician removed the sandbags one by one with the participant performing one session of walking back and forth each time (Figure 1G). This process was repeated until all of the sandbags were removed. Then, the participant walked in place for 15 s without a sandbag (Figure 1H). After the walking session, the participant checked for any signs of discomfort or adverse events (AEs). The needles were removed with the participant standing and bending the lower back region by 15°. PL-MSAT was administered for approximately 10 min at each intervention session.

### 2.8. Criteria for Premature Termination and Dropout

The study was terminated early for the participants due to the following reasons:

(1)Violation of inclusion/exclusion criteria;(2)The participant was diagnosed with a disease that may affect the evaluation of the study outcome, which was not detected in the screening stage prior to the commencement of the study intervention schedule;(3)The participant or their legal representative requested early termination or the participant withdrew consent to participate in the study;(4)Violation of the study protocol by an investigator or participant;(5)The participant was lost to follow-up;(6)Other reasons for which the participant was deemed not appropriate to continue with study participation, according to the judgment of the physician responsible.

### 2.9. Outcome Measures

Outcome measurements were conducted at each timepoint during the study period; the baseline was defined as the timepoint before the intervention on Day 2 of hospitalization (V2-1), and the primary end point was defined as the timepoint after completion of the three sessions of PL-MSAT (V4-2), the 4th day of admission. To ensure consistency in the measurements of outcomes, they were conducted by KMDs with training in the test methods.

#### 2.9.1. Primary Outcome

The primary outcome was the between-group difference in the LBP NRS scores. NRS is a pain scale used to convert the subjective pain intensity experienced by a participant into objective numeric values. A participant was asked to report the severity of the pain felt at the time of measurement by selecting a whole number between 0 (no pain) and 10 (the worst pain imaginable).

#### 2.9.2. Secondary Outcomes

Secondary outcomes included the following: VAS for LBP; leg pain; Oswestry disability index (ODI); lumbar active ROM (aROM); EuroQol 5-Dimension (EQ-5D); EuroQol Visual Analogue Scale (EQ-VAS); Short Form-12 health survey version 2 (SF-12 v2); Patient Global Impression of Change (PGIC); Post-Traumatic Stress Disorder Checklist for DSM-5 (PCL-5-K); AEs; and drug consumption.

Additionally, the primary (NRS of LBP) and secondary (NRS of leg pain, ODI, EQ-5D, SF-12, PGIC, PCL-5-K, AEs, and drug consumption) outcomes were assessed at 1 and 3 months after the date of discharge.

##### Pain

1.Visual analog scale (VAS) for low back pain (LBP)

The VAS records the severity of pain experienced by a participant on a 100 mm horizontal line marked with the descriptor “no pain” at one end and “the most severe imaginable pain” at the other [22]. A participant indicated the severity of LBP as a point on the line.

2.Leg pain

Each participant was surveyed for the presence/absence of radiating pain in the leg.

##### Disability

1.Oswestry Disability Index (ODI)

The lumbar dysfunction of the participants was evaluated using the ODI questionnaire [23]. The ODI questionnaire measured how a participant’s low back disability affected their ability to manage everyday life by dividing the total score by the number of questions answered and obtaining the average value. It is a 10-item questionnaire, and each item is scored from 0 to 5 points; the higher the total score, the more severe the disability of the participant. In this study, a certified and validated Korean version of the ODI questionnaire was used [24].

2.Lumbar active range of motion (aROM)

The maximum lumbar aROM that the participant can move (for six different movement directions (flexion, extension, right lateral flexion, left lateral flexion, right rotation, and left rotation)) without pain was measured.

##### Quality of Life

1.EuroQol 5-Dimension (EQ-5D)

EQ-5D was used to assess five dimensions of a patient’s current health status (mobility, self-care, usual activities, pain/discomfort, and anxiety/depression). Each question is rated on a 5-level scale (Level 1: no problems; Level 2: slight problems; Level 3: moderate problems; Level 4: severe problems; and Level 5: unable to solve the problems).

2.EuroQol Visual Analogue Scale (EQ-VAS)

EQ-VAS aims to serve as an instrument in which patients can express how good or bad their health statuses are. It uses a 100 mm long vertical line showing a scale with values between 100 (best imaginable health) and 0 (worst imaginable health). The patients drew a single line to indicate how good or bad their health statuses were on the day [25].

3.Short Form-12 health survey version 2 (SF-12 v2)

SF-12 v2 consists of 12 items across 8 domains (physical functioning, role-physical, bodily pain, general health, vitality, social functioning, role-emotional, and mental health). The higher scores represent better HRQoL. Kim et al. [26] included 1000 Koreans and verified the reliability and validity of the Korean version of the SF-12.

##### Patient Global Impression of Change (PGIC)

PGIC was used for the subjective and self-reported assessment of a participant’s level of improvement using a 7-point scale (1, very much improved; 2, much improved; 3, minimally improved; 4, no change; 5, minimally worse; 6, much worse; and 7, very much worse) [27].

##### Post-Traumatic Stress Disorder Checklist for DSM-5 (PCL-5-K)

The PCL-5-K is one of the most commonly used self-report measures for assessing post-traumatic stress disorder (PTSD). It was used in this study to measure the degree of PTSD from the trauma factor of road TAs in participants who underwent the accidents. The Korean version of the checklist, which was translated and verified for its reliability and validity by Kim et al., was used [28].

##### Adverse Events (AEs)

AEs refer to any unfavorable and untoward signs, such as abnormalities in the results of laboratory tests, symptoms, or diseases that manifest after the treatment during the study period. The definition of AEs includes events that have no causal relationship with the treatment. In the study, information on AEs was collected through symptoms that were self-reported by participants, and the observations of investigators were recorded without exception. Incidences of AEs suspected of being related to the treatment, according to the investigator’s clinical judgment, were analyzed for abnormal laboratory results and serious AEs (SAEs). The collected safety data were summarized according to the purposes. All SAEs were described in narratives.

For the assessment of causality between the treatment and the AEs, a scale with six categories, created by the World Health Organization-Uppsala Monitoring Center causality assessment system, was used (1 = definitely related, 2 = probably related, 3 = possibly related, 4 = probably not related, 5 = definitely not related, and 6 = unknown). The severity of all the reported AEs was classified into 3 categories according to Spilker’s criteria, as follows: mild (1), involving symptoms requiring no additional treatment with no functional disruption to the participant’s normal activities of daily living (ADLs); moderate (2), involving symptoms causing a significant functional disruption to the participant’s normal ADLs, which may require treatments and disappear over time when additional treatment is applied; and severe (3), involving symptoms that require immediate advanced treatment due to their severity, resulting in sequelae.

##### Drug Consumption

The types, dosages, and regimens of the medications that the participants took during the study period were checked during each visit. In terms of the types of medications, all drugs that had been prescribed for the treatment of the current disease or those that the participants took because of the symptoms related to this study were checked. Besides the medications, the frequency of the treatments for other physical therapy, injections, etc., was recorded.

### 2.10. Data Collection and Management

A Microsoft Excel case report form (CRF) was used. Data entered into Excel CRF were locked after the data cleaning process, and access was blocked for all investigators except for the person in charge of the data.

### 2.11. Statistical Analysis

#### 2.11.1. Analysis Set

Both modified Intention-To-Treat (mITT) and Per-Protocol (PP) analyses were performed, and mITT was set as the primary analysis. PP analysis was performed for participants who underwent treatment sessions no less than three times up to the primary end point.

#### 2.11.2. Missing Data

To handle missing data, multiple imputation was performed for the main analysis set. For sensitivity analysis, mixed models repeated measures (MMRM) and last observation carried forward (LOCF) were used. For multiple imputation, baseline covariates included random allocation, sex, and age in the imputation model, and other covariates were added according to correlations. Predictive mean matching was used for the imputation of missing data. Markov chain Monte Carlo method was used for the estimation process, and 20 imputation sets were generated. Mice package version 3.15.0 was used for missing data imputation.

#### 2.11.3. Baseline Characteristics Analysis

The sociodemographic characteristics of the participants participating in the study were evaluated for each group. Continuous variables are expressed as mean (standard deviation) or median (quartile). Between-group comparisons of continuous and categorical variables were performed using independent *t*-test and chi-squared test, respectively.

#### 2.11.4. Analysis for Outcome Measures

Differences in the change from the baseline for the primary outcome for each group were analyzed. An analysis of covariance (ANCOVA) was used as the primary analysis method. A baseline of each outcome and PCL-5-K were measured a day before the start of the treatment, after the enrolled participants were included as covariates. Additionally, an analysis using a linear mixed model, which included subjects as random effects, was performed.

#### 2.11.5. Secondary Analysis

As a secondary analysis, first, the difference in the area under the curve (AUC) of the NRS and VAS scores up to the primary end point (Day 4) was compared using an independent *t*-test. Next, a survival analysis was performed, considering a decrease of more than 50% of pain from the baseline as an event. The probability of incidence was compared using a Kaplan–Meier graph and a log-rank test, and the hazard ratios were estimated through Cox-regression.

#### 2.11.6. Subgroup Analysis

A subgroup analysis was performed according to participant characteristics, such as sex, age, body mass index (BMI), days from onset, lumbar disc herniation, NRS for back and leg pain, and ODI scores. All statistical analyses were performed using R version 4.3.0 (R Foundation for Statistical Computing, Vienna, Austria), and significance was defined as a 2-sided *p* < 0.05.

### 2.12. Confidentiality

In this study, the management of the personal information of the study participants was undertaken according to rigorous standards under the supervision of the Institutional Review Board (IRB) to ensure confidentiality and data protection. All data collected from study participants who gave their consent to participate in the study were anonymized, and if any data needed to be shared with other institutions, coded information was provided to ensure the protection of personal information.

### 2.13. Ancillary and Post-Trial Care

In the event of the worsening of LBP or unforeseen AEs during the period of study participation, leading to the clinical judgment of the investigators that an individual’s participation is no longer appropriate, the participant was excluded from further study participation. Further, measures were taken for the participant to receive the examinations and medical care that were deemed to be appropriate for treatment according to the standard treatment guidelines. Additionally, information of emergency contacts was collected so that the participants could contact the principal investigator or subinvestigators when they had any questions or experienced medical problems or study-related diseases during the study period.

### 2.14. Potential Biases

Participants and physicians were not blinded. Because PL-MSAT requires more effort and time, the physician’s attention and participant’s expectations may have a positive effects on the effectiveness of PL-MSAT. To reduce bias, an independent researcher unrelated to the treatment served as assessor and remained blind. 

## 3. Results

### 3.1. Participants Flow

The recruitment and follow-up of the study participants are displayed in Figure 2. A total of 104 participants underwent the screening process from July 2021 to June 2022, and all of them met the eligibility criteria and were enrolled in the study without dropouts. The enrolled participants (N = 104) were randomized into the PL-MSAT (N = 52) and control (N = 52) groups. The first study participant was enrolled on July 16, 2021, and the final date of the follow-up visit for the participants was 26 September 2022.

Two participants and one participant from the PL-MSAT group and control group, respectively, withdrew their consent for study participation before the first intervention and were excluded from the analysis. Therefore, 50 and 51 participants in the PL-MSAT and control groups, respectively, underwent the intended treatment and were included in the mITT analysis.

In the PL-MSAT group, five participants were discharged before the completion of the full intervention schedule, resulting in 45 participants completing the procedure. Of the 50 participants included in the mITT analysis, 49 (one withdrew consent during the intervention) participants participated in the telephone follow-up interviews at 1 and 3 months after discharge, respectively. In the control group, 8 participants were discharged before the completion of the full intervention schedule, resulting in 43 participants completing the procedure. In the follow-up at 1 and 3 months after discharge, 48 participants (two participants did not respond to the call, and one participant withdrew consent) participated in both telephone interviews (Figure 2).

### 3.2. Baseline Characteristics

The baseline characteristics of the participants are outlined in Table 1. The mean ages were 43.9 ± 14.0 and 43.3 ± 12.5 years in the PL-MSAT and control groups, respectively. No significant between-group differences were observed for alcohol consumption and smoking in terms of the male/female ratio, heights, body weights, BMIs, or current lifestyles. For the types of TAs, those who had out-of-car TA accounted for 47 (94.0%) and 45 (88.2%) of the PL-MSAT and control groups, respectively, thereby showing a very high percentage of pedestrian accidents in both groups. The days from onset were 3.7 ± 1.9 and 3.7 ± 2.0 days in the PL-MSAT and control groups, respectively, indicating no significant differences between the two groups in terms of TA-related characteristics. 

The numbers of participants diagnosed with lumbar HIVD were 24 and 19 in the PL-MSAT and control groups, respectively, showing similarity between the two groups. The status of lumbar HIVD could not be determined for 18 and 20 participants in the PL-MSAT and control groups, respectively, due to the absence of an in-depth examination in the past or post-occurrence of TA. In addition, the number of participants diagnosed with hyperlipidemia, hypertension, and diabetes was similar in the two groups.

The baseline NRS scores for LBP after TA were 5.7 ± 0.8 and 5.5 ± 0.8 points in the PL-MSAT and control groups, respectively, and the number of participants with leg pain was 16 in both groups; the reported NRS scores for leg pain were 4.8 ± 1.7 and 5.1 ± 1.0, respectively, showing similar values in the two groups. The control group had a higher PCL-5-K score than the PL-MSAT group, even though the group assignment was based on randomization (*p* = 0.049), and the adjusted values of PCL-5-K were applied in all subsequent statistical analyses. Apart from the PCL-5-K score, no significant between-group differences were observed for VAS, ROM, EQ-VAS, EQ-5-D, ODI, and SF-12.

The hospitalization periods for the two groups are presented in Table 2. The total hospitalization periods of the PL-MSAT and control groups were 9.0 ± 3.8 days and 8.9 ± 3.6 days, respectively.

#### 3.2.1. Primary Outcome

The results of the outcome measurements and the between-group differences for each timepoint of measurement are presented in Table 3. Both groups exhibited decreasing NRS scores for LBP. At V4-2, which is the primary end point, the mean NRS scores for LBP were 3.67 (3.44–3.90) and 4.44 (4.20–4.68) in the PL-MSAT and control groups, respectively, indicating significantly lower NRS scores for LBP in the PL-MSAT group by 0.77 (0.44–1.11) (*p* < 0.001). The Cohen’s d was 0.304, indicating a small effect size.

Significant between-group differences in the mean LBP NRS score were observed at various timepoints of outcome assessment, specifically at V2-2, V3-2, V4-1, V4-2, and at discharge; the largest difference in scores was observed at V4-2. Moreover, no significant between-group differences were observed at V3-1, 1 month after discharge, or 3 months after discharge (Table 3).

#### 3.2.2. Secondary Outcomes

##### Pain

Visual analog scale (VAS) for low back pain (LBP)

The mean VAS score for LBP in the two groups showed a trend similar to that of the NRS score. Both groups exhibited a trend of decreasing VAS scores for LBP. At V4-2, the mean VAS scores for LBP were 36.74 (34.30–39.17) and 44.16 (41.70–46.63) in the PL-MSAT and control groups, respectively, indicating a significantly lower VAS score for LBP in the PL-MSAT group by 7.43 (3.93–10.93) (*p* < 0.001). 

For each timepoint of the outcome assessment, significant between-group differences were observed in the mean LBP VAS scores at V2-2, V3-2, V4-1, V4-2, and at discharge; the largest difference in scores was observed at V4-2 (Table 3).

2.Leg pain

No significant between-group differences were observed in the number of participants who complained of radiating pain in the leg at each timepoint from V2-1 to V5, but the number of participants with leg pain was smaller in the PL-MSAT group than that in the control group at V6 and V7 (Table 4). 

**Table 3 healthcare-11-02939-t003:** Comparison of pain outcomes at each measuring point between the PL-MSAT and control groups.

		V2-2	V3-1	V3-2	V4-1	V4-2	D/C	1M	3M
NRS low back pain	PL-MSAT group	5.10 (4.95–5.26)	4.86 (4.65–5.08)	4.55 (4.31–4.79)	4.05 (3.83–4.27)	3.67 (3.44–3.90)	3.20 (2.96–3.44)	2.18 (1.86–2.49)	1.91 (1.63–2.19)
	Control group	5.51 (5.35–5.66)	5.08 (4.87–5.29)	5.03 (4.79–5.27)	4.48 (4.25–4.70)	4.44 (4.20–4.68)	3.56 (3.33–3.80)	2.34 (2.02–2.65)	1.84 (1.55–2.12)
	Difference	0.41 (0.18–0.63)	0.22 (−0.09 to 0.53)	0.48 (0.14–0.82)	0.43 (0.10–0.75)	0.77 (0.44–1.11)	0.36 (0.02–0.70)	0.16 (−0.29 to 0.62)	−0.07 (−0.48 to 0.33)
	*p* value	<0.001 ***	0.166	0.006 **	0.010 **	<0.001 ***	0.038 *	0.473	0.715
VAS low back pain	PL-MSAT group	52.21 (50.38–54.04)	49.59 (47.52–51.66)	45.96 (43.62–48.30)	41.44 (39.12–43.76)	36.74 (34.30–39.17)	31.73 (29.37–34.09)		
	Control group	56.99 (55.18–58.81)	52.27 (50.20–54.34)	51.09 (48.71–53.46)	45.84 (43.51–48.16)	44.16 (41.70–46.63)	35.19 (32.89–37.50)		
	Difference	4.78 (2.16–7.40)	2.68 (−0.30 to 5.66)	5.12 (1.75–8.49)	4.40 (1.06–7.74)	7.43 (3.93–10.93)	3.47 (0.11–6.82)		
	*p* value	<0.001 ***	0.077	0.003 **	0.010 *	<0.001 ***	0.043 *		

Notes: Outcomes were analyzed according to the intention-to-treat principle, and missing data were imputed with multiple imputations. The dashes indicate that outcome measurements were not administered. The outcome measurements at the 14-day follow-up were excluded because they were similar to the discharge outcomes. The mean lengths of stay in the PL-MSAT and control groups were 8.73 ± 3.84 and 8.41 ± 3.91 days, respectively. Five and six participants in the PL-MSAT and control groups, respectively, were discharged before treatment completion. The differences between PL-MSAT and control groups are shown as the mean and 95% confidential interval. Analysis of covariance was performed to calculate the differences and *p*-values. The covariates included each baseline of each outcome and PCL-5-K. The values are presented with 95% confidence interval. * *p* < 0.05; ** *p* < 0.01; *** *p* < 0.001. Abbreviations: V, visit; D/C, discharge; M, month; PL-MSAT, progressive loading–motion style acupuncture treatment; NRS, numeric rating scale; VAS, visual analog scale.

**Table 4 healthcare-11-02939-t004:** Number of patients with leg pain.

	PL-MSAT Group	Control Group
V2-1 (%)	16 (32)	16 (31.4)
V2-2 (%)	16 (32)	16 (31.4)
V3-1 (%)	16 (33.3)	16 (33.3)
V3-2 (%)	16 (33.3)	16 (33.3)
V4-1 (%)	15 (33.3)	15 (34.9)
V4-2 (%)	15 (33.3)	15 (34.9)
V5 (%)	15 (30.6)	16 (32)
V6 (%)	11 (22.4)	15 (31.2)
V7 (%)	10 (20.4)	12 (25)

Notes: The values are presented as number (%). Abbreviations: V, visit; PL-MSAT, progressive loading–motion style acupuncture treatment.

##### Disability

Oswestry Disability Index (ODI)

The ODI showed an overall improvement with time during the study period; however, no significant between-group differences were observed (Appendix A).

2.Lumbar active range of motion (aROM) via physical examination

The ROM for flexion and extension showed higher values in the PL-MSAT group than those in the control group throughout the hospitalization period. At V4-2, the PL-MSAT group exhibited a significantly higher ROM of flexion by −5.31 (−8.15 to −2.48) and extension by −2.09 (−3.39 to −0.80) than the control group. The ROM in the other movements was similar during the hospitalization of the participants, but significant differences were observed at the time of discharge (Appendix A and Figure 3).

##### Quality of Life: EuroQol 5-Dimension (EQ-5D), EuroQol Visual Analogue Scale (EQ-VAS), and Short Form-12 Health Survey Version 2 (SF-12 v2)

The EQ-5D and SF-12 v2 improved during the study period; however, there were no significant between-group differences (Appendix A).

##### Patient Global Impression of Change (PGIC)

The PGIC score of the PL-MSAT group was lower than that of the control group by 0.39 (−0.63 to −0.15) at V2-2, a timepoint immediately after the first session of PL-MSAT. Over time, the between-group difference decreased, and at 3 months after discharge, the PL-MSAT group showed a significantly higher PGIC score than the control group by 0.40 (0.06 to 0.75) (Appendix A).

##### Post-Traumatic Stress Disorder Checklist for DSM-5 (PCL-5-K)

The PCL-5-K score showed an improvement according to the follow-ups. However, no significant differences were observed between the groups (Appendix A).

3.Adverse events (AEs)

AEs were reported in six and four participants in the PL-MSAT and control groups, respectively. One participant in the PL-MSAT group had an AE that was determined to be related to the interventions that were administered during the study period. The reported AE was a nervous system disorder; the participant complained of a mild headache and dizziness, recovering within a day without any additional treatment. Regarding the SAEs, one case of mild hematochezia was reported in the control group; the participant recovered without any additional treatment. The hematochezia was determined to have no causal relationship with the study treatment. 

4.Drug consumption

During the study period, three (6.0%) and two (3.9%) participants in the PL-MSAT and control groups, respectively, took medications related to the symptoms after TAs. Among these five participants, one participant was administered an injection with an analgesic effect once at the study site, and four participants were administered injection to relieve their LBP while visiting another hospital during the treatment period. 

#### 3.2.3. Area under the Curve (AUC)

In the analysis of the cumulative values of each outcome using the AUC, the NRS score was significantly lower in the PL-MSAT group than in the control group by 0.96 (−1.48 to −0.44), and the VAS score was lower by 10.35 (−15.79 to −4.91) (Table 5).

#### 3.2.4. Survival Analysis

A survival analysis was performed based on the definition of recovery as a decrease in the NRS score by ≥50%. The median times to recover measured within 91 days were 41 days (36–43) after randomization in the PL-MSAT group and 41 days (40–46) after randomization in the control group. The values of the log-rank test statistic and hazard ratio were 0.20 and 1.28 (0.89–1.96), respectively, at 91 days, showing no statistically significant between-group differences (Table 6).

#### 3.2.5. Subgroup Analysis

In order to determine the presence of interaction effects according to the participants’ characteristics, a sensitivity analysis was performed for the subgroups. The analysis confirmed that no interaction effects were observed according to the characteristics of the participants, such as sex, age, BMI, days from onset, presence of lumbar disc herniation, NRS of LBP, leg pain, and ODI (Figure 4).

## 4. Discussion

The study results demonstrated that for participants with acute LBP after road accidents, PL-MSAT with IKMT was more effective for pain reduction and improvement in ROM values than treatment with IKMT alone. Both the PL-MSAT plus IKMT combined therapy and the IKMT monotherapy were effective in reducing LBP and improving the ROM and functional outcomes; however, the combination of IKMT with PL-MSAT particularly contributed to the fast relief of pain and an improved ROM.

Regarding the trend of the LBP NRS scores, the NRS scores at V2-2 (the primary outcome) were significantly lower in the PL-MSAT group than in the control group. However, at the timepoint before the second session of intervention (V3-1), no between-group differences were observed. Throughout the treatment duration, the NRS scores were significantly lower in the PL-MSAT group than in the control group at the timepoints after the second session of intervention (V3-2) and before and after the third session (V4-1 and V4-2). The between-group difference in the NRS score was the greatest at V4-2. In addition, the inter-group change in the NRS score in the PL-MSAT group was 1.1 ± 0.12, which is higher than the minimally clinically important difference of 0.9. Therefore, we may infer that PL-MSAT causes a clinically significant improvement in low back pain [29]. This result indicates that combining PL-MSAT with IKMT results in a greater improvement in pain outcomes than IKMT alone. In addition, the effect of PL-MSAT in maintaining the achieved therapeutic effect was superior to that of IKMT monotherapy from the second session of intervention. This difference in effectiveness increased further when the participants underwent the sessions three times or more. At discharge, the NRS scores were significantly lower in the PL-MSAT group than in the control group; the difference was reduced compared to the previous measurements. The reasons for the trend of similar outcomes between the two groups at discharge can be explained by the fact that, in usual clinical practice, the discharge of a participant is determined at the time of a noticeable improvement in pain compared with the pain intensity measured at admission. No significant between-group differences were observed in the NRS scores from the two follow-ups conducted at 1 and 3 months post-discharge. The trend in the VAS scores was similar to that in the NRS scores.

As discussed above, a combination of PL-MSAT with IKMT resulted in a faster improvement in pain outcomes at the early stage of treatment compared with that in the IKMT monotherapy control group. However, no significant between-group differences were observed in the long-term follow-up of the disease. This result can be explained based on the observations in this study. An AUC analysis examining the outcome changes at the primary endpoint showed a significantly superior improvement in the outcomes for the NRS and VAS scores in the PL-MSAT group compared with the control group. Conversely, the survival analysis examining outcome changes over the entire study period showed no significant between-group differences for improvements in the NRS scores.

Furthermore, when PL-MSAT was administered, the ROM at flexion and extension showed better outcomes than the control group during the entire treatment period. The values of the ROM for movements in other directions showed the largest difference at the time of discharge. This result demonstrates that PL-MSAT is also effective for ROM improvement.

In addition, both the NRS and ROM scores showed immediate improvements after the PL-MSAT session. When compared with previous studies, an RCT that used MSAT to treat participants with acute LBP with severe disability [12], as well as another that applied trapezius MSAT for participants with acute whiplash injury [19], reported immediate improvements in the NRS scores and ROM values after the MSAT intervention. These findings are consistent with our observations.

Apart from the outcome measures NRS, VAS, and ROM, the secondary outcomes, ODI, EQ-5D, SF-12 v2, and PCL-5-K, showed overall improvements over time with the treatment sessions, with no significant between-group differences. Therefore, the combination of IKMT with PL-MSAT did not result in significant improvements in disability, quality of life, and PTSD symptoms compared with IKMT monotherapy. Notably, the PGIC score was lower than that of the control group immediately after the first PL-MSAT session at V2-2. This may have been because, as PL-MSAT involves a combination of acupuncture and exercise therapy, the patients’ first experience of a PL-MSAT session may have been challenging depending on the characteristics of the individual participants. Thus, any cases with relatively low PGIC scores may have been due to the slight stiffness and fatigue in the lower back region immediately after the PL-MSAT session. However, as the treatment sessions continued, the participants became accustomed to the PL-MSAT, resulting in slightly higher PGIC scores in the PL-MSAT group. This finding suggests that in the early stages of PL-MSAT treatment, more caution needs to be exercised so participants can familiarize themselves with the treatment process.

Based on disease severity, all 10 AE cases were classified as mild. Among these cases, one was related to the study intervention, in which the participant complained of mild nervous system disorders with headache and dizziness. The symptoms improved without any special treatment. PL-MSAT is a treatment method involving the application of a vertical load to a participant using sandbags, followed by the participant being prompted to walk in place or over a distance. During this process, a participant may experience slight fatigue, dizziness, and headache depending on individual characteristics; however, the severity of these symptoms is mild, and the symptoms improve with a short rest period and require no special treatment. Thus, PL-MSAT does not pose a significant risk to participant safety, and its combination with IKMT may provide a relatively safe treatment option. In a previous RCT investigating the use of trapezius MSAT for participants with acute whiplash injuries, 20 out of 97 participants reported AEs, all of which were mild and did not require special treatment [19].

Further, the influence of sex, age, BMI, days from onset, presence of lumbar disc herniation, NRS of LBP, leg pain, or ODI on the improvement was evaluated. However, no significant interactions between the variables and improvements were observed.

The mechanism of the immediate pain relief effect and functional improvement of PL-MSAT have yet to be elucidated. However, previous studies suggest that the associated mechanism is the promotion of pain reduction through progressive loading, the analgesic effect of acupuncture, and changes in pain perception. If the distal acupuncture points are strongly stimulated during the MSAT session, the central nervous system is internally activated, inducing “diffuse noxious inhibitory controls,” which promotes the release of endorphins and leads to analgesic effects [30]. During the PL-MSAT session, in addition to the procedure of the MSAT session, isometric exercises are performed while additional progressive loading is applied. This leads to reduced pain sensitivity due to the exercises, which immediately reduces pain and promotes muscle strengthening [31,32]. The results of this study confirmed that the NRS scores and ROM were the two main outcomes that showed significant improvements. Regarding additional mechanisms underlying the therapeutic effects of MSAT, a complex interplay between the effect of promoting self-regulating ability through a combination of scalp acupuncture and movement therapy, ending the negative cycle of pain, and the role of joint movement in preventing connective tissue adhesion has been discussed [33].

In the PL-MSAT session, patients can overcome the fear of pain that may arise during movements under the guidance of physicians/KMDs. As reported previously, addressing the fear of pain can positively affect the improvement of pain and functional outcomes in participants [34,35]. In addition, the fear avoidance belief is one of the risk factors of the transition to chronic LBP in patients with acute LBP [36]. The treatment may facilitate the process of rehabilitation and prevent the acute LBP from developing into a chronic condition.

### 4.1. Strengths and Limitations

This study has some strengths. First, we evaluated the daily outcome measures of various aspects, including pain, disability, and quality of life in patients with acute LBP. By observing the consecutive changes in the acute phase and continuing long-term follow-ups, we could understand the full prognosis of acute LBP after TA. Second, this was the first randomized controlled trial that investigated the effectiveness and safety of PL-MSAT. Third, since PL-MSAT is a novel technique combining MSAT and exercise, this study may provide evidence for expanding the developed mode of acupuncture-based rehabilitation.

However, this study has some limitations. First, the blinding of the participants and physicians was not performed due to the evident and significant difference in treatment methods between the two groups. To ensure objectivity, assessor blinding was applied in this study; however, the possibility of a placebo effect cannot be ruled out in this setting. In addition, NRS was reported directly by the participant as the primary outcome, which means that the actual outcome assessor was not blinded. Hence, the bias could not be completely resolved by blinding the independent assessor.

Second, this study used IKMT, which is commonly used for the treatment of LBP after TAs in current clinical practice [37], as a control intervention. Thus, the effect size may have been small compared with cases where a different treatment method was used as the control intervention. In addition, in the PL-MSAT group, IKMT was performed in combination with PL-MSAT; thus, the possibility of synergistic effects cannot be ruled out, making it difficult to identify the independent effects of PL-MSAT.

Third, 60 participants had mild LBP of NRS 5 [38] at baseline. Since several participants with mild symptoms were included as study participants, significant effects with meaningful clinical benefits were not easily observed because the change from the baseline ROM after the treatment was not large. In this situation, confirming the interaction between the baseline NRS and the treatment effect from the subgroup analysis was difficult.

In addition, only three sessions of treatments were performed for the participants with mild symptoms, and the follow-up was performed 1–3 months after discharge. Thus, at the timepoints of the long-term follow-up, both the treatment effect of PL-MSAT and the natural course of the acute mild LBP may have affected the outcomes. Consequently, it would be difficult to observe the differences in therapeutic effects compared to the outcomes of IKMT monotherapy. Therefore, from our clinical perspective, we suggest additional studies performing PL-MSAT in more sessions for patients with moderate to severe pain who require more active treatments to relieve pain and prevent chronicity. With these studies, the long-term effects of this intervention can be confirmed more clearly.

Furthermore, at the timepoints of the long-term follow-up, evaluating whether the acute LBP developed into chronic LBP or a relapse status would provide useful insights.

Finally, although the pain outcomes showed a faster improvement in the PL-MSAT group, no difference was observed in the hospitalization period between the two groups. The timing of discharge may have been unaffected by the treatment method, because no significant between-group differences were observed in the functional outcomes. Therefore, future RCTs should include the independent evaluation of the clinical efficacy of PL-MSAT for patients with acute LBP of moderate severity, with a limited number of treatment methods administered in parallel.

### 4.2. Further Implications

PL-MSAT could be applied to other diseases in which MSAT and progressive loading exercises have been used separately. For instance, Huh [20] reported that MSAT improved pain and disability in patients with lumbar HIVD. Steele [39] reported that exercise using a high load, low volume, and low frequency can cause the healing and regeneration of discs. By combining these previous studies, PL-MSAT may be considered for patients with lumbar HIVD.

Korean Medicine Clinical Practice Guideline for Traffic Injuries includes the recommendations of MSAT for post-TA neck pain. However, no evidence regarding post-TA LBP has been obtained yet. For future clinical guidelines, this study will provide evidence of PL-MSAT as a developed method of MSAT. When making decisions in the medical field, PL-MSAT could be considered when the patient has a high level of fear avoidance beliefs.

For the economic implications, as pain relief was rapid in PL-MSAT, some patients may be able to speed up their recovery to daily life and reduce productivity loss in the early stages of a traffic accident.

This study contributes significantly to the literature because the results demonstrate the effectiveness and safety of the PL-MSAT for immediate pain relief and the improvement of functional outcomes for acute LBP caused by road TAs. The findings indicate that in the treatment of acute LBP after a road accident, combining PL-MSAT with other treatment modes may provide clinical benefits and therapeutic effects.

## 5. Conclusions

The combination of PL-MSAT with IKMT may provide better clinical outcomes in reducing pain and increasing the range of motion than a treatment with IKMT alone. However, there were no significant differences in disability, quality of life, and PTSD symptoms. It is expected that treatment including PL-MSAT will provide clinical benefits for patients with acute LBP, and further studies will be needed to establish this efficacy with high-quality evidence.

## 6. Protocol Version

The study protocol version is 1.6 (2022. 05. 03). Major revisions of the study protocol and other changes made after this paper will be updated at the trial registration site.

## Figures and Tables

**Figure 1 healthcare-11-02939-f001:**
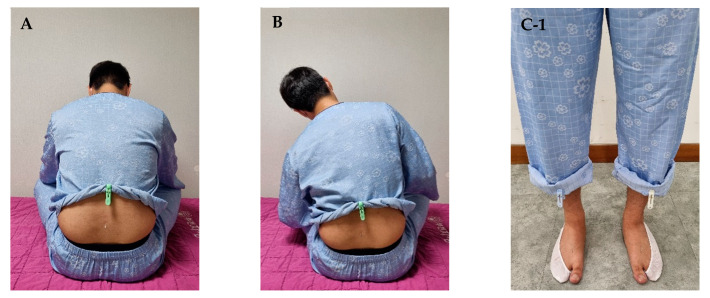
Progressive loading–motion style acupuncture treatment. (**A**) Perpendicular needling into GV4 (命門, Myeong-Mun) or GV3 (腰陽關, Yoyang-gwan) is performed by the physician, but approximately 3 mm of the body of the needle is not inserted. (**B**) With the patient in a cross-legged position with the needle partially inserted, the physician gently rocks the participant’s upper body 15 times on either side. (**C-1**,**C-2**) The participant is asked to stand with the partially inserted needle, and another needle is inserted into bilateral LR2 (行間, Haeng-gan). (**D**) The participant walks in place for 15 s, followed by walking forward for a certain distance. (**E**) The participant is asked to fold their arms, atop which the physician places one sandbag. The participant then walks back and forth twice in a straight line over a distance of 10 m. **(F**) The physician adds one more sandbag on top until the participant has three sandbags, and the participant walks back and forth twice between each bag. (**G**) The physician removes the sandbags one by one, and the participant walks back and forth once between each bag. (**H**) The participant walks in place for 15 s without a sandbag, and then the needles are removed.

**Figure 2 healthcare-11-02939-f002:**
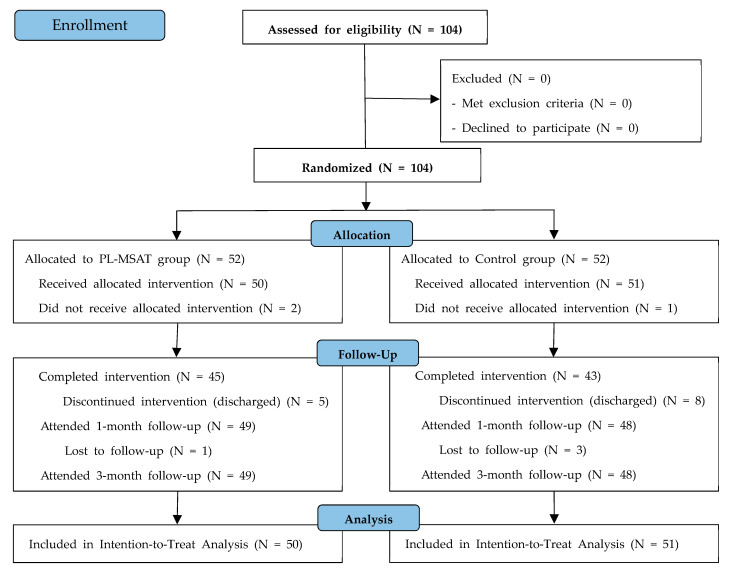
Flow chart of the participant enrollment process.

**Figure 3 healthcare-11-02939-f003:**
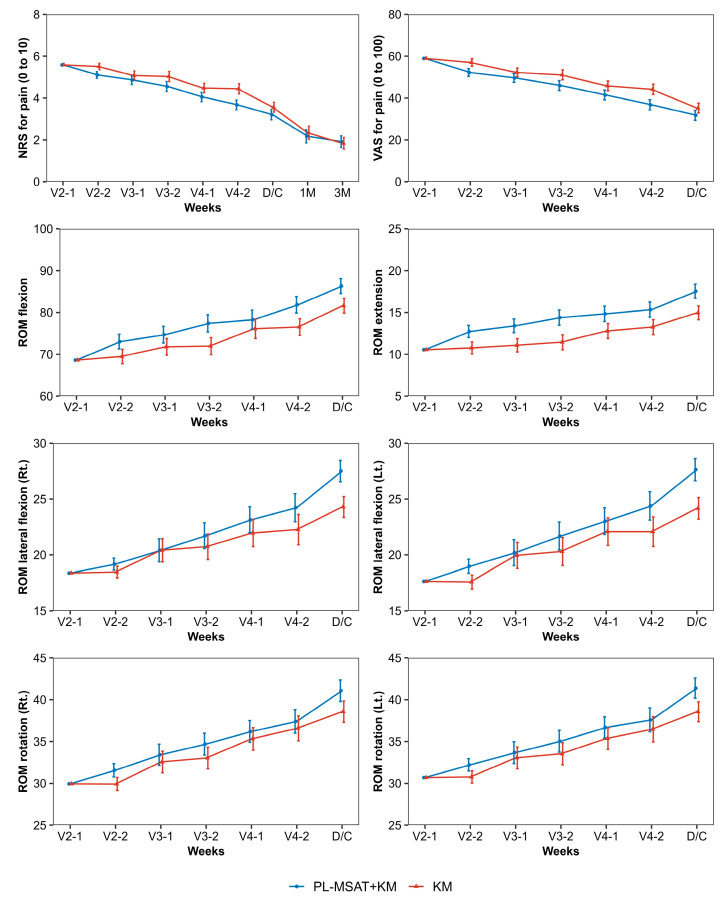
Comparison of NRS and range of motion between the PL-MSAT and control groups. All figures show curves for the treatment period (baseline to discharge), with 95% confidence intervals represented by vertical bars. The plots and 95% confidence intervals are presented with least squares estimates and their 95% confidence intervals. From the timepoint of visit 2-2 (after one session of PL-MSAT), the NRS and VAS scores were lower in the PL-MSAT group than in the control group; for all types of ROM, the measurements were larger in the PL-MSAT group than in the control group. Abbreviations: V, visit; D/C, discharge; M, month; PL-MSAT, progressive loading–motion style acupuncture treatment; IKMT, integrative Korean medicine treatment; NRS, numeric rating scale; VAS, visual analog scale; ROM, range of motion; Rt., right; Lt., left.

**Figure 4 healthcare-11-02939-f004:**
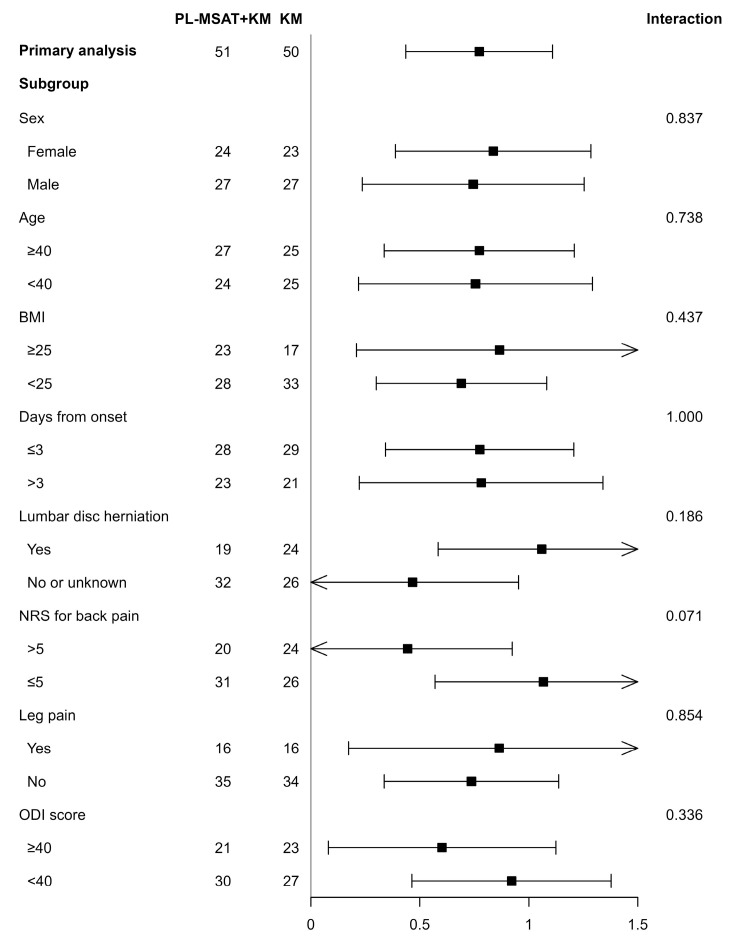
Subgroup analysis. Abbreviations: PL-MSAT, progressive loading–motion style acupuncture treatment; BMI, body mass index; NRS, numeric rating scale; ODI, Oswestry disability index.

**Table 1 healthcare-11-02939-t001:** Baseline characteristics of the participants.

		PL-MSAT Group	Control Group	*p*-Value
		(n = 50)	(n = 51)	
Sex (%)	Female	23 (46.0)	24 (47.1)	1.000
	Male	27 (54.0)	27 (52.9)	
Age (year)		43.9 ± 14.0	43.3 ± 12.5	0.825
Height (cm)		168.3 ± 8.5	166.7 ± 8.2	0.341
Weight (kg)		69.2 ± 12.4	70.2 ± 15.3	0.721
BMI (kg/m^2^)		24.3 ± 2.8	25.2 ± 4.7	0.255
Current alcohol consumption (%)	Yes	19 (38.0)	18 (35.3)	0.940
	No	31 (62.0)	33 (64.7)	
Current smoking (%)	Yes	14 (28.0)	18 (35.3)	0.566
	No	36 (72.0)	33 (64.7)	
HL	Yes	7 (14.0)	6 (11.8)	0.969
	No	43 (86.0)	45 (88.2)	
HT	Yes	6 (12.0)	5 (9.8)	0.972
	No	44 (88.0)	46 (90.2)	
DM	Yes	5 (10.0)	4 (7.8)	0.741
	No	45 (90.0)	47 (92.2)	
HIVD	Yes	24 (48.0)	19 (37.3)	0.478
	No	8 (16.0)	12 (23.5)	
	Unknown	18 (36.0)	20 (39.2)	
Days from onset (mean (SD))		3.7 ± 1.9	3.7 ± 2.0	0.948
In-car TA	Yes	3 (6.0)	6 (11.8)	0.487
	No	47 (94.0)	45 (88.2)	
Expectation—PL-MSAT group		6.2 ± 1.2	5.8 ± 1.0	0.078
Expectation—control group		5.0 ± 0.9	4.8 ± 0.8	0.209
Other medications at baseline (%)	Yes	0 (0.0)	1 (2.0)	1.000
	No	50 (100.0)	50 (98.0)	
NRS score for low back pain		5.7 ± 0.8	5.5 ± 0.8	0.335
Existence of leg pain (%)	Yes	16 (32.0)	16 (31.4)	1.000
	No	34 (68.0)	35 (68.6)	
NRS score for leg pain		4.8 ± 1.7	5.1 ± 1	0.456
VAS score for low back pain		59.7 ± 8.2	58.2 ± 7.5	0.337
EQ-VAS		45.6 ± 15.8	44.6 ± 16.9	0.762
ODI		38.5 ± 13.3	37.2 ± 14.0	0.633
ROM	Flexion	68.2 ± 10.8	69.0 ± 9.2	0.682
	Extension	10.8 ± 3.6	10.3 ± 2.1	0.385
	Right lateral flexion	18.6 ± 3.9	18.1 ± 3.7	0.545
	Left lateral flexion	18.0 ± 3.8	17.3 ± 4.3	0.356
	Right rotation	30.4 ± 5.3	29.5 ± 5.6	0.415
	Left rotation	30.5 ± 5.4	30.9 ± 5.5	0.723
PCL-5-K		18.4 ± 12.7	23.5 ± 13.0	0.049
EQ-5D		0.69 ± 0.14	0.69 ± 0.12	0.944
SF-12 (PCS)		40.1 ± 8.0	37.7 ± 7.5	0.122
SF-12 (MCS)		45.1 ± 12.0	42.1 ± 11.4	0.197

Notes: Data are expressed as mean ± standard deviation (SD) or number (%). Between-group comparisons of continuous and categorical variables were performed using an independent *t*-test and chi-squared test or Fisher’s exact test, respectively. Abbreviations: PL-MSAT, progressive loading–motion style acupuncture treatment; BMI, body mass index; HL, hyperlipidemia; HT, hypertension; DM, diabetes mellitus; HIVD, herniated intervertebral disc; TA, traffic accident; NRS, numeric rating scale; VAS, visual analogue scale; EQ-VAS, EuroQol visual analogue scale; ODI, Oswestry disability index; ROM, range of motion; PCL-5-K, Post-Traumatic Stress Disorder Checklist for DSM-5; EQ-5D, Quality of Life EuroQol 5-Dimension; SF-12, Short Form-12 health survey version 2; PCS, physical component summary; MCS, mental component summary.

**Table 2 healthcare-11-02939-t002:** Hospitalization period.

	Hospitalization Period	
	Mean ± SD	Median (IQR)
PL-MSAT group	9.0 ± 3.8	6 (8–13)
Control group	8.9 ± 3.6	6 (8–13)

Abbreviations: SD, standard deviation; IQR, interquartile range; PL-MSAT, progressive loading-motion style acupuncture treatment.

**Table 5 healthcare-11-02939-t005:** Areas under the curves for the outcomes.

	PL-MSAT Group	Control Group	Difference	*p*-Value
NRS	11.60 (11.23–11.96)	12.56 (12.19–12.92)	−0.96 (−1.48 to −0.44)	<0.001 ***
VAS	118.52 (114.76–122.27)	128.86 (125.07–132.65)	−10.35 (−15.79 to −4.91)	<0.001 ***

Notes: The area under the curve was calculated using the trapezoidal rule. The differences between the two groups were analyzed using analysis of covariance. Missing values were imputed using multiple imputations. All values are presented with least square estimates and their 95% confidence intervals. *** *p* < 0.001. Abbreviations: PL-MSAT, progressive loading–motion style acupuncture treatment; NRS, numeric rating scale; VAS, visual analog scale.

**Table 6 healthcare-11-02939-t006:** Survival analysis.

	Event	Median Survival Time (Days)	Log-Rank	HR
PL-MSAT group	44	41 (36–43)	0.20	1.28 (0.83–1.96)
Control group	39	41 (40–46)		

Notes: Survival analysis was performed with the event of “recovery of participants with ≥50% reduction in NRS of LBP”. Abbreviations: PL-MSAT, progressive loading–motion style acupuncture treatment; NRS, numeric rating scale; HR, hazard ratio.

## Data Availability

This RCT was registered on Clinicaltrials.gov (clinical trials ID: NCT05493007). Further information on the hospital and investigators can be found on the trial registration site.

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
