# Peer review of "Effectiveness and Safety of Progressive Loading–Motion Style Acupuncture Treatment for Acute Low Back Pain after Traffic Accidents: A Randomized Controlled Trial"

_healthcare, 2023, doi:10.3390/healthcare11222939_

Round 1

Reviewer 1 Report

Comments and Suggestions for Authors

Dear Authors,

Thank you for submitting your manuscript titled " Efficacy and Safety of Progressive Loading Motion Style Acupuncture Treatment (PL-MSAT) on Inpatients with Acute Low Back Pain After Traffic Accidents: A Randomized Controlled Trial" for consideration in the Healthcare journal.

Upon thorough review, I am pleased to comment on the high quality of your research and the relevance of the topic addressed. The methodological design and the results you've presented offer a valuable contribution to the field and hold potential influence in clinical practice, especially in treating acute low back pain post-road accidents.

However, in spite of the undeniable strengths of your work, I believe there are several areas that could benefit from further revision to enhance the manuscript's suitability for publication in Healthcare. These suggestions are offered in the spirit of refining the paper to ensure its message is communicated in the clearest and most effective manner possible.

1. Comment

Your manuscript requires expert English editing.

2. Comment in reference to the Abstract

2.1 Introduction. While the background sets the purpose of the study, it might benefit from an additional sentence that contextualizes the significance of the issue. For instance, how prevalent is post-road traffic accident LBP? Such context could help set the relevance of the study.

2.2 Methodology Details.

Could you provide more details on how randomization was achieved? Were stratification or block methods used?

The justification or determination of sample size is not clear in the abstract. Consider elucidating this for better clarity.

2.3 Study Cohort.

The mention of "104 randomized patients" followed by "50 PL-MSAT and 51 control group patients" amounts to 101, not 104. This discrepancy needs clarification. What became of the other 3 patients?

2.4 Ensure that terminology is consistent.

For instance, if referring to low back pain as "LBP" initially, maintain that abbreviation throughout the abstract.

2.5 Results.

It would be helpful to include measures of variability (such as standard deviation or standard error) alongside the means presented.

Apart from the NRS scores, were there other secondary outcomes? If so, what were they and what were the findings?

2.6 Conclusion.

The claim about "increasing range of motion" is seen in the conclusions but isn't mentioned in the results. If it's a key finding, it should be included and quantified in the results section.

2.7 Study Design Details.

It might be useful to specify if patients were randomized in a 1:1 ratio between the treatment and control groups.

3 Comment in reference to the Introduction

3.1 Relevance and Contextualization.

The initial lines effectively contextualize the global significance of road traffic injuries. However, it would be helpful if the introduction also provided a brief overview of the global prevalence or impact of LBP, not just in the context of road traffic accidents.

3.2 Specific to Korea.

The data on Korea, while informative, seems densely packed. Consider breaking it into more digestible sections or presenting some of the statistical data in a table format for clarity.

3.3 While the introduction covers all necessary points, it could benefit from clearer structuring. For instance, starting with global statistics, moving to national (Korea) data, and then focusing on specific treatments.

3.4 There is some repetition in the information, especially regarding LBP post-traffic accidents. It might be beneficial to condense these details.

3.5 Ensure that transitions between sections are smooth. For example, after presenting data on traffic injuries in Korea, the transition to treatments like acupuncture should be smoother.

3.6 The passage on MSAT and PL-MSAT is technical. While it's well-explained, the authors might consider simplifying it or providing more context for those unfamiliar with these techniques.

3.7 Consistency.

At times, the text refers to "motion style acupuncture" and at other times to "Motion style acupuncture therapy (MSAT)". It would be helpful to be consistent in terminology after the first mention.

4. Comment in reference to the Materials and Methods

4.1 Elimination of the "Objectives" Subsection.

I observed that the initial subsection titled "Objectives" might be redundant as the objectives have already been addressed towards the end of the introduction. I suggest considering the removal of this subsection to avoid repetition and enhance the cohesiveness of the document.

In a clinical trial, it's imperative that the Materials and Methods section starts with the "Study Design" to set a clear context. From there, it's advisable to structure the content into well-defined subsections, addressing aspects such as the type of trial, primary objective, setting, and duration. Subsequently, details on inclusion and exclusion criteria, randomization and allocation methods, blinding, applied interventions, studied variables, follow-up procedures, and the statistical analysis plan should be laid out. This structured approach will enhance clarity and adhere to standard reporting practices.

4.2 Clarity and Structure.

I recommend reconsidering the formatting of the "Materials and Methods" section to align more closely with the PRISMA structure. This will ensure each critical component of the study is addressed in a clear and systematic manner.

4.3 It's essential to keep the terminology consistent. For instance, "patient" and "participant" are used interchangeably. While this might be acceptable in some contexts, it's essential to ensure that the terminology remains consistent to avoid potential confusion.

4.4 Sentence Structure.

There are some long sentences which might be split for clarity. For instance:

"AEs refer to any unfavorable and untoward signs (e.g., abnormality in the results of laboratory tests), symptoms, or diseases that manifest after the treatment during the study period; and the definition of AEs include the events that have no causal relationship with the treatment."

This could be split into:

"AEs refer to any unfavorable and untoward signs, such as abnormalities in the results of laboratory tests, symptoms, or diseases that manifest after the treatment during the study period. The definition of AEs includes events that have no causal relationship with the treatment."

4.5 Clarity on Measures.

When introducing measures, be consistent in explaining what they mean. Some measures are thoroughly explained (like EQ-5D-5L), while others are only briefly introduced (like PGIC).

4.6 Following the CONSORT guidelines, I suggest adding a subsection in the methodology on "Potential Biases." It is essential to discuss possible biases and the strategies used to manage them, which will bring greater rigor and transparency to your manuscript.

4.7 In the 'Outcome Measures' section, I recommend restructuring based on the specific outcome variables (e.g., lower back pain, disability, quality of life) rather than the measurement instruments. For each variable, subsequently detail the instruments used. This approach will make the section more intuitive for readers. Please consider making these adjustments for clarity.

4.8 Upon reviewing the 'Methodology' section of your manuscript, I find it to be overly extensive, with this expansiveness evident across its various subsections. Particularly, sections like statistical analysis appear too broad, while the 'Eligibility' section, organized as a list of terms, lacks fluidity. I strongly suggest revisiting and refining these subsections to eliminate redundancies and ensure a more coherent and logical presentation. Condensing and streamlining this content will significantly enhance readability and clarity for your audience.

5. Comment in reference to the Results

5.1 Upon review of the 'Results' section in your manuscript, I advise a restructuring for improved clarity and coherence. Instead of organizing the section based on the measurement instruments used, it would be more logical and reader-friendly to divide it according to the specific variables analyzed. As such, please consider restructuring Table 3 to reflect the analysis of these variables accordingly. Moreover, I noticed that data pertaining to primary variables are presented within the subsection for secondary variables, which could lead to confusion. It's imperative to ensure a clear delineation between primary and secondary outcomes in both the text and tables. Please adjust these sections to provide a more streamlined presentation of your findings.

5.2 I've pinpointed a key area that requires your attention: the presentation and analysis of effect size and both intra and inter-group changes. Effect size not only quantifies the magnitude of observed change or difference but also provides deeper insight into the clinical relevance of your findings, beyond mere statistical significance. It's pivotal for researchers and practitioners to gauge not just if there was a change, but the magnitude of that change.

Additionally, to enhance clarity and coherence in your 'Results' section, I recommend reorganizing this section into subsections based on the variables analyzed. Then, within each subsection, present the results obtained from each measurement instrument employed. This layout will make it easier for readers to follow and understand your findings.

6 Comment in reference to the Discussion and Conclusions

6.1 After carefully reviewing the discussion section of your study on PL-MSAT, I would like to suggest greater attention to the clinically significant changes, in addition to the statistically relevant ones. It would be valuable to determine whether changes in PL-MSAT have a practical and tangible impact on patients' lives and if they justify its clinical adoption over other interventions. Furthermore, expanding on the relevance of these findings in everyday clinical practice would be enriching. Lastly, a deeper exploration of how PL-MSAT compares with other treatments from a practical standpoint would be appropriate. Essentially, an approach that not only emphasizes statistical significance but also its applicability in the real clinical setting would add value to your discussion.

6.2 I would recommend that within the discussion section, you create specific subsections for "Strengths and Limitations". This structure would offer better organization and clarity for readers. Additionally, emphasizing the study's strengths in its own subsection would provide a balance, highlighting what supports the validity of your findings and the aspects that might influence them.

6.3 Regarding "Applicability in Different Populations": It's crucial to consider the generalization of your findings across various populations or demographic contexts. Given that your study focuses on a specific population, delving deeper into how PL-MSAT might be relevant or applicable in other groups would be invaluable. Expanding on this would allow for a deeper understanding of the scope and limitation of your findings and could also guide future research in diverse populations.

6.4 Concerning "Future Implications": It's vital to provide a more detailed insight into how your study might influence future clinical guidelines and decision-making in the medical field. Additionally, outlining potential future research that your work suggests or inspires would be beneficial. This would not only enrich your current discussion but also provide a clear path for researchers and clinicians interested in furthering this domain.

6.5 Regarding the "Interaction of Factors": Given that various variables or factors might be at play, it would be valuable to discuss their potential interactions and how they could collectively influence the results. Such a discussion can shed light on the intricacies of your findings and offer readers a more comprehensive understanding of the nuances involved.

6.6 On the subject of "Economic Implications": Beyond the evident clinical benefits, it is imperative to delve deeper into the economic ramifications of your study. For instance, is PL-MSAT more cost-effective than other treatments? Moreover, considering the broader economic landscape, how would the widespread adoption of PL-MSAT influence healthcare budgets? Are there potential savings in the long run, or are there hidden costs that need to be factored in? Addressing these economic aspects can provide a holistic view of PL-MSAT's potential impact, making your discussion more robust and well-rounded.

6.7 The conclusions presented in your clinical trial are largely clear and direct. However, there is room for more specificity and depth. I noticed that some secondary variables analyzed seem to have been omitted in your conclusions. It is essential that these outcomes are detailed more specifically to provide a full understanding. For instance, instead of mentioning "superior outcomes", it would be beneficial to specify exactly which outcomes or measures showed the most significant improvements. While recognizing the study's strengths is crucial, it would also be prudent to briefly mention the main limitations to provide a balanced view. Lastly, integrating a concise comparison with existing literature could further contextualize your findings within the broader scope of research in the field. Such inclusion would not only strengthen your conclusions but also provide clear direction for future research.

7 Comment in reference to the References

7.1 It's essential that your bibliographic references adhere to the specific guidelines of the journal you plan to submit to. Please review and format your citations and references according to the journal's style guide to ensure consistency and compliance with their editorial requirements. Doing so will streamline the review and publication process for your manuscript.

I trust that these comments and suggestions will be received in the constructive spirit in which they are offered. I look forward to your revision and the opportunity to consider your revised work for publication in Healthcare.

Comments on the Quality of English Language

Minor editing of English language required.

Reviewer 2 Report

Comments and Suggestions for Authors

The authors presented a randomized control study assessing patients with low back pain post-road accidents, contrasting the effects of progressive loading motion style acupuncture therapy (PL-MSAT) with Integrative Korean Medicine Treatment (IKMT) that incorporates acupuncture therapy. Their findings indicated an earlier alleviation of pain in the intervention group, although no significant difference was observed upon extended follow-up. While the trial's registration in ClinicalTrials.gov and its adherence to the study design lend credibility to their methodology, I'd like to point out some considerations.

Acupuncture therapy, specifically PL-MSAT, might not be widely recognized outside East Asia, yet this study could captivate clinicians practicing these therapies. However, the study's major shortcoming lies in its blinding aspect. While the authors argue that the outcome assessor was blinded, the primary outcome – a numeric rating scale of pain – is inherently patient-reported. Consequently, the patient is, in essence, the actual outcome assessor. This effectively means that no genuine blinding took place for the primary outcome. The authors do acknowledge the blinding issue in their limitations, but it would be prudent to explicitly emphasize the absence of blinding concerning the primary outcome.

Furthermore, the abstract doesn't allude to the long-term outcomes which showed no significant differences. Even if the long-term outcome is designated as a secondary outcome in the protocol, neglecting to mention it can be perceived as introducing bias, potentially being construed as a "spin". It's imperative for the authors to incorporate long-term outcome findings within the abstract for a more comprehensive representation.

Lastly, the study's conclusion appears overly assertive. Stating, "The combination of PL-MSAT with IKMT showed significantly better clinical outcomes in reducing pain and increasing range of motion than treatment with IKMT alone," may overstate the results. Moreover, if the authors opt to highlight "range of motion" (a secondary outcome) in the abstract, it is only fitting that other secondary outcomes, regardless of their statistical significance, should also be addressed.

Minor suggestions for corrections are as follows:

Line 122: The definition of "traffic accidents" should be further clarified. Does it encompass all scenarios such as pedestrian-vehicle collisions, vehicle-to-vehicle accidents, or any traffic-related incident? Should all types of traffic accidents be considered, major accidents likely would have been ruled out naturally. This is because patients requiring hospitalization for issues other than back pain (like bone fractures) would have been excluded. Additionally, the statement, "Patients requiring inpatient care due to acute LBP" is ambiguous. More explicit criteria detailing the prerequisites for admission due to acute LBP are necessary for better comprehension.

Line 283: The primary outcome's timeframe was defined as 4 days in the study's protocol. This timeframe should be mentioned in this line for clarity and consistency.

Line 718: According to inclusion criterion 3), only patients with an NRS score 5 or more were included. The term "mild" in this context is unclear. Does it refer to patients with an NRS score of 4 or less? The specific NRS values that constitute "mild" should be stated. Moreover, if the authors believe that "additional studies specifically targeting patients with moderate or severe symptoms are warranted," they might consider a supplementary subgroup analysis for such patients to reinforce their hypothesis.

Round 2

Reviewer 1 Report

Comments and Suggestions for Authors

Dear authors,

I have carefully reviewed your manuscript titled "Effectiveness and Safety of Progressive Loading Motion Style Acupuncture Treatment (PL-MSAT) on Inpatients with Acute Low Back Pain After Traffic Accidents: A Randomized Controlled Trial". Firstly, I would like to commend you for the effort put into this study and for the improvements made to the manuscript since the last review.

However, there are still a number of errors and inconsistencies that need to be addressed before considering the manuscript for publication. It is crucial that these areas be rectified to ensure the integrity and clarity of the study.

Concise Title: Your current title is extensive and might benefit from being made more succinct without losing the core message. I suggest revisiting the title to trim it down, focusing on key words that convey the aim and methodology of your study in a more straightforward manner.

Use of MeSH terms: To enhance the visibility and discoverability of your manuscript in databases and search systems, I recommend identifying and employing terms from the Medical Subject Headings (MeSH) that align with the main thrust of your study. This will not only aid in the indexing of your work but also ensure that potential readers can easily find your study when conducting related searches.

Please condense the abstract to a maximum of 250 words, ensuring it provides a concise overview without omitting essential information.

Selection Criteria Wording: I've noticed that in the selection criteria section, there's consistent repetition of terms, notably "patients". For clarity and conciseness, I would advise you to draft this section in paragraph format rather than as a list. In doing so, you can consolidate information and eliminate unnecessary redundancies, providing a clear and concise description of your selection criteria.

Upon reviewing your manuscript, I noted that abbreviations were frequently used in the titles and subtitles. For the sake of clarity and to ensure that all readers, especially those who might not be familiar with specific abbreviations, can understand the context, I strongly recommend that you use the full term accompanied by its abbreviation in parentheses for the first instance in titles and subtitles. For example, instead of using "EQ-VAS", it would be clearer to present it as "EuroQol Visual Analogue Scale (EQ-VAS)". This would provide immediate clarity while also introducing the abbreviation for subsequent use in the text.

I observed that this occurs in several subtitles throughout the manuscript. Addressing this consistently will greatly enhance the readability and comprehensibility of your paper.

It's essential that your bibliographic references adhere to the specific guidelines of the journal you plan to submit to. Please review and format your citations and references according to the journal's style guide to ensure consistency and compliance with their editorial requirements. Doing so will streamline the review and publication process for your manuscript.

To ensure clarity and comprehensibility for all readers, I kindly request that you verify that all abbreviations are expanded and defined upon their first occurrence in the text. This practice is essential to maintain a clear and coherent presentation throughout your manuscript.

Comments on the Quality of English Language

Moderate editing of English language required.
